# Narrative Messages and the Use of Emotional Appeals on Endometriosis Screening Intention: The Mediating Role of Positive Affect

**DOI:** 10.3390/ijerph20136209

**Published:** 2023-06-23

**Authors:** Allison Worsdale, Jiaying Liu

**Affiliations:** 1Department of Communication Studies, University of Georgia, Athens, GA 30602, USA; 2Department of Communication, University of California Santa Barbara, Santa Barbara, CA 93106, USA; jiaying-liu@ucsb.edu

**Keywords:** endometriosis, narrative messages, emotional appeals, positive affect, screening intention, young adult women

## Abstract

Endometriosis affects around 10% of women globally, yet the awareness and screening rates for this condition are relatively low. Utilizing an online survey-based experiment with a sample of 18–30-year-old young women (*N* = 326), this study aimed to investigate the efficacy of narrative messages vs. non-narrative messages for promoting endometriosis screening intention, as well as to evaluate the effectiveness of hope appeal vs. fear appeal in narrative messages. The study also examined the potential mediating mechanisms through self-efficacy and positive affect responses that may help elucidate the effect of emotional appeals on behavioral intentions, while taking into account an individual’s readiness to change. Findings indicated that narrative and non-narrative messages did not produce significantly different screening intentions. Compared to the use of fear appeal, the hope appeal in narrative messages predicted a higher level of positive affect responses, which was associated with increased endometriosis screening intentions. Individual difference in readiness to engage in endometriosis screening was not found to be a significant moderator. These results have implications for future research utilizing hope appeals in narrative health messaging. The observed significant mediational pathway through positive affect advances understanding of positive discrete emotions as facilitators to health-related cognition and behavior changes.

## 1. Introduction

Endometriosis, a condition in which the lining of the uterus grows outside the uterus, is a chronic illness characterized by severe and painful menstruation, pain during intercourse, and long-term health impacts such as depression and infertility. This disease affects around 1 in 10 women of reproductive age globally [1], but lack of awareness, as well as normalization of women’s menstrual pain, prevents women from getting screened and receiving proper treatment [2]. A UK survey found that 62% of women aged 16–24 are unaware of what it is [3]. This results in a significant delay of diagnosis and treatment, or no diagnosis at all, due to never being screened for endometriosis. Health communication efforts that can effectively promote endometriosis screening awareness and intention are crucially needed.

While numerous health message strategies exist, narratives set themselves apart in terms of creating heightened personal relevance of the target behavior in the audience [4]. Previous research focusing on women’s health has found success in the use of narratives, specifically regarding screening and prevention behaviors [5,6,7]. Many narrative studies have explored women’s screening behaviors in regard to breast and cervical cancer [1,8]; however, less is known about the effectiveness of narrative messages on endometriosis screenings. Emotional appeals in narrative messages have been demonstrated to be impactful in promoting behavioral shifts, as emotions hold a significant sway over people’s decision-making processes [9,10]. Some emotional appeals, such as fear appeal, have been widely studied due to their capacity to elicit strong emotional responses that are often successful at motivating people to take immediate actions; the use of other emotional appeals in narrative messages, such as hope appeal, has received less scholarly attention, despite their potential to inspire people in positive ways [11]. This disparity in research attention highlights the need to study the effects of hope appeals and the ways in which they can be used to create more powerful and impactful narrative health messages.

The present study attempts to fill these gaps by proposing and testing how narratives may promote the behavioral intentions associated with endometriosis screening. The study will examine the differences between the presentation of endometriosis screening through narrative versus non-narrative evidence, as well as narratives using hope and fear appeals. The study further tests self-efficacy and positive emotional response as mechanisms that may explain the impact of narratives on behavioral intention. Additionally, an individual’s readiness to change is examined as a moderator of these potential mediational mechanisms.

## 2. Literature Review

### 2.1. Using Narrative Approaches in Health Research

Hinyard and Kreuter define narratives as “any cohesive and coherent story with an identifiable beginning, middle, and end that provides information about scene, characters, and conflict; raises unanswered questions or unresolved conflict; and provides resolution” [12] (p. 778). Previous health communication literature employing narratives in health messages has proven effective across a wide array of topics [13,14]. In these studies, researchers have discovered that narratives can be powerful drivers of behavioral change as they evoke emotions and imagination, offering a relatable context to comprehend health-related information and leading to higher levels of engagement. Conversely, non-narrative messages—i.e., messages that do not tell a story and do not have any lead character who engages the reader with their thoughts, feelings, and experiences and that present information through a didactic format or statistical data—are less likely to capture attention, sustain interest, or enable effective message internalization. For example, De Wit et al. [15] found that individuals had lower behavioral intentions to get vaccinated when presented with information about the Hepatitis B virus in a didactic format that made no reference to a specific character. 

Shen et al. [14] found that narratives advocating detection and prevention behaviors were more effective than those promoting cessation behaviors. Therefore, narratives may be able to serve as an effective tool to promote endometriosis screening and treatment, given that it is a chronic disease that necessitates early detection. Further studies examining the effect of narratives on behavioral intention changes observed that participants exposed to the narrative conditions had higher intentions to receive health screenings compared to those in the non-narrative conditions [16,17]. These studies shared a common focus of positively promoting the recommended behavior by portraying characters in the messages as engaging in the behavior and illustrating the desired outcomes that ensue, as opposed to describing the negative outcomes following non-engagement of the recommended behavior. This aligns with De Graaf et al.’s [9] systematic review that found that narratives advocating health promotion behaviors with positive outcomes were associated with more significant changes in behavioral intentions than negatively valanced narratives. In view of this, the present study aims to examine the effect of a narrative message promoting endometriosis detection with positive outcomes on people’s screening intentions, as compared to that of a non-narrative message. We hypothesize the following:

**H1.** *Information focusing on endometriosis detection presented in a narrative format will be more effective in increasing individuals’ behavioral intentions to get an endometriosis screening, compared to the same information presented in a non-narrative format*.

### 2.2. The Use of Hope Appeals in Narratives

Persuasive, narrative messaging often increases in effectiveness when emotional responses are elicited in audience members [18]. In de Graaf et al. [9], researchers found emotional appeals to be an important content characteristic that increased persuasion in several studies using emotional adjectives and descriptions. Given the important role of emotional appeals in narrative persuasion, extensive research has been conducted to examine how intended affective responses can be elicited by the emotional appeals used in narratives [19,20]. Most of such research focuses heavily on elicitation of negative emotions such as fear, anger, guilt, etc., which has shown success in changing individuals’ health behaviors [21,22,23]. Historically, fear has been the primary emotion studied by scholars when attempting to engage individuals and shift their intentions or behaviors [11]. Fear has shown to be most effective when certain levels of efficacy and threat are met. Specifically, when individuals perceive a high level of efficacy, they are more inclined to act in order to mitigate a threat, such as a potential negative health consequence [23]. However, the effectiveness of fear can wane, especially if individuals do not believe they can effectively change their behavior to avoid the threat [11]. In response to the limitations of negative emotional appeals, especially fear appeals, the use of positive emotional appeals has been proposed as an alternative and effective tool to promote health behavior changes [11]. Studies comparing the effectiveness of positive and negative emotional appeals in health messages, however, have yielded mixed results. While some studies have suggested that positive emotions can be more effective in motivating behavior change compared to negative ones [24,25], others have found minimal differences between the two types of emotional appeals in their impacts on factors such as risk perceptions [26]. Despite these mixed results, researchers focusing on positive emotional appeals have found success in changing people’s behavioral intentions through incorporating hope appeals in narrative messages [13,16,17,27]. Moreover, some studies have observed a significant mediational pathway through positive affect when examining the effect of hope narratives on health outcomes [28,29].

Hope appeals have recently attracted increasing scholarly attention due to their powerful impacts on an individual’s health intentions and behaviors when incorporated into persuasive narrative messages [11,24]. Hope is a vital emotion characterized by an optimistic outlook about the immediate or long-term future, and has been theorized to influence motivation, attitudes, and behaviors in meaningful ways, leading to goal achievement [30]. It is the sense that things will get better, that one’s efforts will be rewarded, and that there is a reason to persevere despite challenges and setbacks. Importantly, when attempting to elicit hope within audience members using narratives, Nabi [18] suggests that it first needs to stem from unpleasant circumstances leading to a positive, better future outcome. In relation to the current study, hope appeal may be a particularly relevant and effective tool given the nature of endometriosis screening, i.e., the relief people may experience once they are able to be screened and get treatment for their ailment. 

In discrete emotion literature, hope has often been conceptualized as a singular emotion. However, Myrick and Oliver tried to understand hope from the perspective of mixed emotions or dual emotional arousal to account for its powerful influence. Use of mixed emotional appeals has been found to be more effective in motivating individuals to engage in behavior change than singular emotions [25,31]. Messages invoking solely negative emotions can result in issues such as weaker message effects or unintended negative consequences. However, dual emotion arousal (both negative and positive emotions) can influence post-message behaviors as the emotions elicited can motivate individuals in behavior or belief change while the positive emotions, specifically, can lower post message discomfort and negative consequences [11]. Although hope is often deemed as a positive, singular emotion, its unique underlying cognitive appraisal structures—that implicate both the negative, unsatisfactory elements that exist within our life experiences, threatening perceptions of well-being or goal attainment (such as uncertainty, frustration, fear, despair, etc.), and remedial prospects or positive outcomes (such as solutions, desirability, etc.)—resemble that of the co-occurrence of multiple discrete emotions. 

The great potential of utilizing hope appeals in persuasive messages promoting behavior change may also be understood through the lens of emotional flow [32]. Emotional flow involves one’s emotional experience as it evolves while being exposed to a health message typically marking one or several emotional shifts. Valence of shift can be negative to positive or vice versa, as well as shifts to emotions of similar valence (i.e., fear to anger [31]). Nabi and Green [33] suggested that these emotional shifts can be promoted in audiences as they are designed for individuals to follow character progression as events occur in narratives, especially when the character overcomes adversity. Such shifts can sustain narrative engagement, enhance the emotional intensity experienced, and in turn produce narrative-consistent cognition and behavioral changes [34]. In the current study, showcasing a character overcoming the adversity of dealing with endometriosis and eventually achieving desirable health outcomes may be effective in eliciting the emotion of hope and experiencing shifts from negative emotional states (e.g., feeling anxious, powerless, unable to take action) to positive ones characterized by optimism, excitement, and motivation.

A unique outcome when utilizing hope appeals in narratives is the production of meaningful outcomes [25]. Specifically, the idea behind meaningful emotional outcomes is that these narratives not only can motivate viewers for behavior changes but also aid in the enhancement of subjective well-being and feelings of fulfillment in the audience members that view them [35,36,37]. This is particularly relevant in the behavioral context of the current study; given the painful experiences (e.g., pain during periods, with intercourse, with bowel movements or urination, excessive bleeding, etc.) and anxiety-inducing uncertainty (e.g., potential infertility) associated with the medical condition of endometriosis, it is crucially important that narrative messages are developed not only to focus on changing behaviors but also on improving people’s affective well-being at the same time.

Some scholars have begun exploring the idea of *restorative narratives*, which comprise two key components: strength-based messages and incorporation of meaningful progression to incorporate hope appeals. Strength-based narratives aim at conveying themes of resiliency [31], and the incorporation of meaningful progression highlights the process of overcoming hardship, coping with the implications of one’s current state, and suggesting future positive outcomes in individuals’ cognitions [10]. These two components act together to trigger the affective response of hope within the viewers of the narrative. In other words, restorative narratives centralize on the emotional appeal of hope to influence people’s cognitions and behaviors. Restorative narratives do not create an absence of negative emotion experienced by participants (e.g., fear, anxiety, etc.), but results should indicate higher levels of positive and meaningful emotions than negative emotions. This is due in part to the journey-approach of the narrative wherein individuals are exposed to the negatives of the topic being presented, followed by the positive outcomes that could occur if the participant is motivated by the message. Therefore, the current study proposes to utilize the restorative narrative, with hope appeals incorporated, to promote endometriosis screening and examine the effectiveness of hope appeals against fear appeals which are heavily utilized in narrative persuasive messages.

Specifically, based on the theoretical and empirical evidence reviewed above, we put forth the following hypothesis:

**H2.** 
*Narrative messages incorporating hope appeals will (a) lead to higher self-efficacy to get an endometriosis screening, (b) result in more positive affective responses, and (c) produce higher behavioral intentions to get an endometriosis screening, compared to narrative messages incorporating fear appeals.*


Considering that the unique nature and structure of hope appeals may resemble the co-occurrence of multiple discrete emotions and bear analogy to emotional flows in narratives that ultimately result in positive affect, we hypothesize that hope appeals are more effective than fear appeals in promoting cognitive and behavioral changes that align with the narrative due to the enhanced positive emotional experiences they create. Additionally, prior studies have suggested that the positive outcome depicted in a hope appeal can lead individuals to perceive it as achievable. This perception of attainability can boost their self-confidence in performing the desired behavior, ultimately elevating their level of self-efficacy. Furthermore, considering that individuals’ self-efficacy has been widely recognized as an important antecedent to subsequent behavioral changes with classic behavior change theories [38,39], and empirical research has observed the important mediating role of self-efficacy between emotional appeals and intention changes [5,11,23,27,40]. Thus, we propose the following:

**H3.** *(a) Self-efficacy and (b) positive affect will mediate the relationship between emotional appeals in the narrative format (hope vs. fear) and behavioral intentions to get an endometriosis screening*.

### 2.3. The Potential Moderating Role of Individual Differences in Stage of Change

While narratives can be effective tools in engaging individuals in health behavior change, it is important to identify what stage of change they are in and how this may influence the effectiveness of the message. The Transtheoretical Model of Behavior Change (TTM) construes individual change as a process rather than an event. For example, Prochaska and Velicer [41] suggest that change will happen over time and is not bound by a strict sense of temporality. The core framework of the model includes six stages of change individuals go through when adopting or stopping a behavior: precontemplation, contemplation, preparation, action, maintenance, and termination [42]. Most individuals will follow the sequence of change outlined in the stages; however, the path is not simply linear as many may move through the stages and regress back if they experience a relapse in behavior change.

Given that endometriosis screening is a one-time behavior (i.e., after taking the action of screening, the following stages would not be applicable), the stages of precontemplation, contemplation, and preparation can be used to describe the different phases before one actually takes the action of screening. Specifically, individuals in pre-contemplation refer to those who are unaware of endometriosis or do not plan to get screened for endometriosis; individuals in the contemplation stage refer to those who acknowledge that having an endometriosis screening is important but are not ready or lack confidence to get screened; and individuals in preparation stage refer to those who are getting ready to get screened but have not taken the action yet.

Classic TTM literature has been utilizing the readiness-to-change variable to capture and describe one’s stage of change on a continuum [43,44]. Readiness is conceptualized as a combination of an individual’s perceived importance of a particular health concern or problem and the efficacy to engage in the change [44]. Since the theoretically proposed discrete stages are not necessarily mutually exclusive, the continuous readiness-to-change variable may be a more accurate predictor of one’s level of preparation or engagement in a health behavior. This study will aim to quantify each participant’s readiness to change toward screening for endometriosis and examine how this will impact how the messages will be received and their effectiveness. This leads into the following research question:

**RQ1.** *Does individual difference in readiness to change moderate the mediation relationships proposed in H3*?

## 3. Materials and Method

### 3.1. Participants

Participants were recruited via an undergraduate student research pool at a large Southeastern university. The study included women who were 18–30 years old. The justification for this criterion is that while endometriosis can impact women from age 15–40, it is typically diagnosed in women aged 18–30 [45]. The study also required that eligible participants should self-identify as biological females, and at least experience some symptoms associated with endometriosis (such as menstrual pain, or pain having sex, etc., although not necessarily caused by endometriosis). Women who had been previously screened for endometriosis were excluded. An a priori power analysis, utilizing a small to medium effect size of Cohen’s *f* = 0.20 [46,47,48], indicated that at least 274 participants would be needed to ensure an 80% chance of detecting the medium/small effect size between the three experimental groups as significant at the 5% level, two-tailed [49]. A total of *n* = 326 participants’ data were collected for the study, which were adequate to test the hypotheses.

### 3.2. Study Design and Procedures

The study consisted of a survey-based online experiment. Before the experiment, eligible participants were queried with a set of questions regarding their readiness to undergo endometriosis screening and asked to provide their basic demographic information. Next, participants were assigned to one of three conditions: a non-narrative message condition, a narrative message with fear appeal condition, and a narrative message with hope appeal condition. Participants then answered post-exposure questions measuring their levels of self-efficacy, affective responses, and intentions towards endometriosis screening. Following the post-exposure questions, participants were debriefed and thanked for their participation.

### 3.3. Stimuli

Messages across all conditions were static and embedded in a website mock-up resembling that of BuzzFeed. The narrative conditions featured stories from the perspectives of women being screened for endometriosis. The non-narrative condition featured similar information presented in a non-story format. Across all conditions, despite the presentation format difference, the information contained in each message (i.e., explaining the disease, the symptoms, and how individuals can get screened) was kept equivalent. All message stimuli can be found in the Appendix A.

Stimuli development went through a rigorous procedure. The initial message drafts were first presented to a group of 15 research assistants tasked with providing constructive feedback to improve the message’s readability and overall appearance. The subsequent feedback was then applied to the messages to improve the look and content. The revised messages underwent readability testing and iterative revisions until achieving a similar and appropriate level of readability across three conditions. The finalized messages all fell within sixth to eighth grade readability (scores of 7.5–9.5) on the Automated Readability Index [50]. The messages contained equivalent information to avoid confounding influence and required a similar amount of effort to read each condition. Across three conditions, the texts had similar lengths of around 550 words.

To ensure successful manipulation of the message stimuli, we conducted a pilot study to test if the stimuli were correctly perceived as the message type they were designed as (i.e., narrative vs. non-narrative, fear vs. hope appeals). The test was conducted among *n* = 68 female young adults (18–30 years old, *M* = 19.52, *SD* = 2.15) recruited through the undergraduate research pool at the same institution. Once deemed eligible, participants were then informed that they would be asked to evaluate a health message. The message was randomly chosen from the three message stimuli. After message exposure, as detailed in the next section, participants were asked a series of questions to determine whether they perceived the message types as intended.

### 3.4. Measures

All variables were measured on a 5-point Likert scale from “strongly disagree” to “strongly agree” unless otherwise noted.

#### 3.4.1. Manipulation and Attention Checks

The same two sets of manipulation check questions were used in both the pilot and the main experiment studies. The first set of questions was used to determine whether the narrative versus non-narrative manipulation was successful and was asked of participants in all conditions. Based on Escalas’ [51] 4-item manipulation check, we asked participants to rate how strongly they agreed with statements such as “the message showed the personal evolution of one or more characters.” The other manipulation check was conducted only among those in the narrative conditions, to determine whether participants viewed and perceived the fear appeal and hope appeal narratives as intended. After message exposure, participants in the narrative conditions were asked to what extent they agreed with two different statements related to how the character felt about her disease. The statements consisted of “she was optimistic about her prognosis” and “she was scared about her prognosis”.

In the main experiment study, in addition to manipulation checks, an attention check was included in the post-exposure survey. Following the behavioral intention measure, participants were asked, “People vary in the amount they pay attention to these kinds of surveys. Some take them seriously and read each question, whereas others go very quickly and barely read the questions at all. If you have read this question carefully, please only select strongly disagree.” All participants included in the sample passed the attention check.

#### 3.4.2. Stages of Change

To measure the stage of change a participant is currently in, the pre-exposure questions used the Biener and Abrams’ [52] Contemplation Ladder originally developed for measuring the stages of change in the context of smoking cessation. The 11-item scale was adapted for endometriosis screening in this study. Items included statements such as, “I have no interest in getting screened for endometriosis.” Only the first seven items which were pertinent to the precontemplation, contemplation, and preparation stages were used in the current study. These items were analyzed by calculating an individual’s readiness scores [44]. These scores were calculated by taking the mean values for each stage (precontemplation, contemplation, preparation) and subtracting the precontemplation mean from the summation of the other two stages. This produced the readiness value that was used for analysis (*M* = 1.22, *SD* = 1.95, α = 0.84).

#### 3.4.3. Self-Efficacy

The survey adapted a self-efficacy scale developed by Champion et al. [53] that was designed to measure women’s self-efficacy to obtain mammography. Because the scale was originally designed for women’s screening behaviors, it was appropriate for the current study’s purpose to measure women’s endometriosis screening behavior. The scale consisted of 10 items adapted with specific language regarding endometriosis screening (e.g., “you can find a place to have endometriosis screening”, “you can make an appointment for endometriosis screening” (*M* = 4.20, *SD* = 0.60, α = 0.93)).

#### 3.4.4. Positive Affective Response

Participants’ positive emotional responses were measured using an adapted version of Dillard and Peck’s affective response scale [54]. This scale was previously adapted by Fitzgerald et al. (2019) in the context of restorative narratives to elicit the emotional response of hope. The adapted scale contained four subscales, but the current study only utilized the positive subscale to measure positive affect. Participants were asked to rate how much they felt each of the emotions while reading the message. Items for the positive affect scale included happy, cheerful, joyful, and upbeat measures on a 5-point Likert scale from “not at all” to “very much” (*M* = 1.88, *SD* = 0.86, α = 0.95).

#### 3.4.5. Behavioral Intention

To measure a participant’s endometriosis screening intention, a 3-item measure of intention was adapted from Ratcliff et al. [55]. Example items included, “I want to get screened for endometriosis in the near future” (*M* = 2.83, *SD* = 0.96, α = 0.84).

#### 3.4.6. Demographics

Participants’ age, biological sex, gender, and race/ethnicity were measured. Detailed demographic information is presented in Table 1.

Participants were required to identify as biologically female, however, participants of any gender orientation were able to participate. One participant in this sample identified themselves as female biologically and male as gender orientation.

## 4. Results

### 4.1. Pilot Study for Message Testing

Results indicated that the manipulation of both narrative and non-narrative messages as well as fear and hope appeals were effective. When analyzing the 4-item narrative manipulation scale for narrative versus non-narrative conditions, reliability analysis suggested removing one item from the scale: “*The message had a beginning, middle, and end*”. After removing the item to improve the internal consistency of the scale (*α* = 0.86), we averaged the remaining three items to create a scale variable (*M* = 4.99, *SD* = 1.45) for the manipulation check, with higher scores indicating greater consistency with a narrative message format. A significant difference was observed for non-narrative (*M* = 3.64, *SD* = 1.53) and narrative (*M* = 5.64, *SD* = 0.93) conditions on narrative manipulation, *t*(66) = −6.68, *p* < 0.001. For manipulation of hope and fear appeals, there were significant differences between groups for the hope appeal (*M* = 3.21, *SD* = 1.41) and fear appeal (*M* = 2.64, *SD* = 0.58), for the item, “*She was optimistic about her prognosis,*” *F*(1, 44) = 4.46, *p* = 0.04. For item 2, “*She was scared about her prognosis*”, there was a significant difference between groups for the hope appeal (*M* = 3.33, *SD* = 1.09) and fear appeal (*M* = 3.95, *SD* = 0.79), *F*(1, 44) = 4.84, *p* = 0.03. With these successful manipulation check results, we proceeded using the message stimuli, with minor modifications based on participants’ open-ended responses, in the main study.

### 4.2. Main Study Results

The main study sample included a total of *n* = 326 participants. Demographic characteristics are presented in Table 1. The mean age of the sample was 19 years old (*SD* = 1.15). The majority of the sample was White (77.6%), female-identifying (99.7%), and non-Hispanic White (93.3%). Table 2 describes the mean and standard deviation values for each of the focal variables of the study broken down by condition. Table 3 shows the correlations among the focal variables. All analyses were adjusted for race, ethnicity, and age as covariates.

Manipulations of both narrative and non-narrative messages as well as fear and hope appeals were effective for the main study. There was significant difference in scores for non-narrative (*M* = 2.90, *SD* = 0.09) and narrative (*M* = 4.17, *SD* = 0.03) conditions, *t*(324) =16.33, *p* < 0.001. Additionally, there was a significant difference in scores for fear (*M* = 3.39, *SD* = 0.41) and hope (*M* = 3.50, *SD* = 0.66) conditions, *t*(218) = −1.58, *p* < 0.001.

As summarized in Table 4, the narrative messages (*M* = 2.84, *SD* = 1.01) were not more effective at increasing behavioral intentions to get an endometriosis screening than the non-narrative message (*M* = 2.82, *SD* = 00.87), *F*(1, 321) = 0.07, 95% CI [−0.26, 0.20], *p* = 0.80. H1 was not supported.

For hypothesis testing, H1 predicted that the narrative messages would be more effective at increasing intentions to get an endometriosis screening than the non-narrative message. We tested H1 with an Analysis of Covariance (ANCOVA) analysis with the two-category experimental condition variable (non-narrative vs. narrative) as the independent variable and intention to get an endometriosis screening as the dependent variable. 

H2 predicted that narratives using hope appeals would be more effective at increasing individuals’ self-efficacy, positive affect, and behavioral intention towards getting an endometriosis screening. ANCOVA analyses were performed to analyze the effects of emotional appeal types used in the narrative conditions (i.e., fear vs. hope appeals) on efficacy, positive affect, and intention. As summarized in Table 4, no significant difference in self-efficacy was observed between the fear (*M* = 4.19, *SD* = 0.62) and hope appeal conditions (*M* = 4.15, *SD* = 0.61, *F*(1, 215) = 0.51, 95% CI[−0.10, 0.22], *p* = 0.47. A similar pattern was also observed when intention was examined as a dependent variable. Specifically, individuals in the fear appeal condition (*M* = 2.94, *SD* = 0.89) did not differ significantly compared to those in the hope appeal condition (*M* = 2.75, *SD* = 1.10) in terms of their intention to get an endometriosis screening, *F*(1, 215) = 1.94, 95% CI [−0.08, 0.46], *p* = 0.17. However, when examining the effect of different emotional appeals on positive affect, the findings suggested a significant difference between the two conditions, such that individuals in the hope appeal condition experienced significantly greater positive affect (*M* = 2.30, *SD* = 0.86) compared to those in the fear appeal condition (*M* = 1.71, *SD* = 0.74), *F*(1, 215) = 26.61, 95% CI [−0.79, −0.35], *p* < 0.001. Thus, H2 was partially supported, with H2a and H2c (self-efficacy and intention) not supported, and H2b supported (positive affect).

H3 proposed to examine whether self-efficacy and positive affect would mediate the relationship between types of emotional appeals and intention to get screening. H3 was examined using the PROCESS macro-Model 4 [56,57], to test the proposed mediational pathways between emotional appeals and intention through efficacy and positive affect. The independent variable of condition was coded using the Helmert coding system, with the non-narrative message (condition 1) as the reference group, the fear-based message (condition 2) as value 2, and the hope-based message (condition 3) as value 3 [56]. This allowed for both comparison of the narrative messages and the non-narrative reference group as X1 and the comparison of fear-based and hope-based messages as X2. As shown in Table 4, the mediational pathway through self-efficacy was not significant (Indirect effect = −0.02, 95% CI [−0.09, 0.04]). Therefore, H3a was not supported. When examining H3b, which proposed that positive affect would mediate the relationship between types of emotional appeals and intention, a significant indirect effect (Indirect effect = 0.11; 95% CI [0.01, 0.23]) was observed. Detailed unstandardized path coefficients for the mediation models can be found in the Appendix A (Appendix A). Therefore, H3b was supported.

RQ1 sought to examine the potential moderation effect of individuals’ readiness to change on the mediational pathways tested in H3. For RQ1, two separate moderated mediation analyses (Model 7) in PROCESS were conducted to examine whether the mediational pathways of positive affect and efficacy on intention were moderated by individuals’ readiness to change [56,57]. All the analyses conducted in PROCESS also adjusted for the same set of covariates used in the ANCOVA analyses. The results suggested a non-significant interaction effect between types of emotional appeals and readiness to change on the mediator variable, i.e., self-efficacy (*B* = 0.08, *p* = 0.06). The overall moderated mediation model was also not significant (index of moderated mediation = 0.01, 95% CI [−0.02, 0.04]). Similarly, readiness to change was also not found to moderate the effect of types of emotional appeals on positive affect (*B* = −0.07, *p* = 0.23). The overall moderated mediation model was not supported either (index of moderated mediation = −0.01, 95% CI [−0.04, 0.01]). In summary, individuals’ readiness to change did not moderate the mediational pathways between types of emotional appeals and intention via self-efficacy and positive affect.

## 5. Discussion

Endometriosis is a disease in which uterine tissues grow in areas outside of the uterus, resulting in abdominal pain, pain with intercourse, and potential infertility as well as other symptoms. Screenings involve healthcare providers determining, through a checklist of symptoms and a pelvic exam, if an individual does have endometriosis and further treatment is necessary. Despite the fact that endometriosis affects around 10% of women of reproductive age globally [1], the population remains poorly educated about the disease, the importance of screening, and the corresponding treatment options. Due to the overall low disease literacy, symptoms may be dismissed and diagnosis and treatment may be delayed, leading to exacerbated negative impacts on those affected. Effective health communication and education efforts are crucially needed to change the situation.

Health messages presented in the narrative format provide important information about goals, plans, actions, and outcomes of health/risk behaviors from the perspective of one or more protagonists within a story. A non-trivial amount of prior research has demonstrated their ability to influence emotional response to health messages as well as their effectiveness in shifting individual health outcomes [13,16,17]. On the other hand, there was also evidence supporting the use of expository non-narrative messages to more effectively present health-related facts [25,30]. Increasingly, more research has started to examine the boundary conditions and mechanisms through which narrative format may enhance or undermine persuasion. To understand if and how narrative format may enhance efficacy of health messages in educating women about the disease, and increasing their intention to screen for endometriosis, this study examined the effects of both narrative and non-narrative formats as well as the use of fear and hope appeals in narrative messages in the context of promoting endometriosis screening.

Our results indicated that, in the context of promoting endometriosis screening, narrative messages did not produce a significantly higher screening intention compared to the non-narrative message. This may be explained by the fact that awareness surrounding endometriosis is relatively low; it is an illness that is underrepresented in popular media and frequently misdiagnosed as other ailments [58]. This pattern is also corroborated by our own dataset, a highly educated sample from a university research pool, as about 50% of our participants reported not having any prior knowledge of endometriosis (*n* = 164). Because of this, participants may not be familiar with the symptoms and typical experience an individual with endometriosis goes through. In this situation, the non-narrative informational message may be equally effective since it provides information in a direct format to those with little to no background knowledge of the disease. Previous research has emphasized the success of narrative messages in contexts of well-known illnesses where individuals lack motivation to engage in behaviors such as smoking or sun-damage prevention [13,59]. Additionally, past work has indicated that while narratives can enable people to experience emotions as if the story’s events were real, non-narrative messages or messages employing statistical or informational evidence may have a greater chance of influencing outcomes such as attitude, efficacy, and intention that rely more on rational deliberation or calculation [26,60]. Furthermore, because the narrative messages in the current study only highlighted one individual’s story, this may have given people more reason to believe that they will not necessarily experience the same events themselves, which is in line with previous research using individualized stories in health messages [61]. Future studies could evaluate the effectiveness of narrative versus non-narrative messages between individuals who are familiar with endometriosis and those who are unfamiliar.

When comparing narrative messages containing fear and hope appeals regarding the character’s endometriosis diagnosis, we observed that the two messages did not produce significantly different results in self-efficacy. This finding is not consistent with results from prior studies comparing these emotional appeals in other contexts which observed that hope is often more successful at increasing self-efficacy than fear [11,62,63]. One potential explanation for this finding is that the participants in our sample had relatively low awareness of the disease and, therefore, they may not have been fully aware of what the health behavior of getting screened entails. There was a relatively high mean score for the self-efficacy variable across conditions (*M* = 4.20, *SD* = 0.60), meaning participants across conditions may believe the behavior to be relatively easy to complete. The overall high levels of efficacy beliefs may not provide adequate variance in this variable for the detection of significant differences across conditions. Additionally, positive effects of narrative messages incorporating hope appeals on self-efficacy are often seen when individuals are fully aware of the need to engage in a recommended health behavior or end an undesirable behavior, but they do not believe they are able to do so or perceive barriers preventing them from doing so [11,27]. In these studies, self-efficacy was exhibited for well-known health behaviors (exercise and skin-cancer prevention behavior, respectively) where individuals know that they will benefit from engaging these behaviors. In these situations, narrative messages incorporating hope appeals are well positioned to impact their efficacy to perform or maintain the behaviors when compared to fear narratives. However, in our study, due to the lack of awareness of the disease, participants may need to first be informed of the general information about the disease and accept that they *should* engage in the recommended behavior; in other words, while our messages were helpful in raising awareness, participants were not yet at the stage where sufficient information about the recommended behavior was absorbed for them to start making adequate, differential efficacy evaluations.

When comparing the effects of the two emotional appeals on positive affect, however, the findings suggest that the hope narrative did significantly and positively predict greater positive affect experienced by participants compared to the fear narrative. This is consistent with previous research on incorporating meaningful and hopeful emotions in narratives which found that the narratives incorporating hope appeals did in fact increase positive affect following message exposure [10,11]. In addition, although it was found that hope appeal was not more effective at increasing behavioral intention than fear appeal, findings suggested that the indirect effect of hope appeal on intention through positive affect was significant, such that compared to fear appeal, hope appeal increased the positive affect that individuals experienced, which in turn was associated with increased screening intentions. This finding is in line with previous research emphasizing the mediating effects of positive affect [10,29,60]. Considering that the emotion of hope involves moving from unpleasant circumstances to positive beliefs about future outcomes, this finding is also consistent with previous research on employing mixed emotions in health messages in that researchers found indirect effects of narrative types on behavioral intentions to engage in the targeted health behavior through changes in affective responses [11,25]. These findings highlight the notion that evoking discrete emotions, particularly the mixed emotion of hope, which lands on positive emotional experience and presumably arouses higher levels of emotional intensity due to the transition of the different emotions involved, can facilitate higher intentions to engage in desired health behaviors. This set of findings supports the ongoing shift from elicitation of fear to that of hope in order to influence individuals’ health behavioral intentions and explicates how narrative health messages incorporating hope appeal may exert desirable impacts on intentions by affecting one’s positive affect experience in the context of endometriosis screening.

Finally, individual difference in readiness to engage in endometriosis screening was not found to moderate the proposed mediational pathways. About 50% of participants in the current study identified themselves as being within the precontemplation stage (*n* = 162), which may be partially due to lack of knowledge surrounding the disease. Consistent with prior research on stages of change [64,65], our findings indicate that as an individual’s readiness increased, so did their intention and positive affect (Table 3). However, when testing the moderation effect of readiness on the two mediational pathways, the results did not differ across those scored higher or lower on the readiness variable. It could be that, as most participants were at early stages of change where they had no thoughts of endometriosis or screening for the disease, the mediational pathways worked (in the case of positive affect serving as mediator) and did not work (in the case of self-efficacy serving as the mediator) equally across the whole sample given the negligible differences in their readiness scores. Both lack of knowledge and most of the sample predominantly being in precontemplation stage may explain why there was no moderation effect.

The study is not without limitations. First, the study used a convenience sample as participants were all recruited from a university research pool. Therefore, the findings may not be generalizable to a larger population of women. Future studies should recruit from more representative sources as this will help exhibit differences in health literacy, readiness to change, and education levels that can impact results. Second, much of the sample was in the precontemplation stage on the readiness-to-change scale. This may have impacted the lack of significance of stage of change as a moderator due to the minimal variance in readiness. Future studies may find significance employing quota sampling methods wherein there is equal distribution of each stage of change. Third, a “no message” control group was not included. The current study utilized an informational message as the non-narrative condition which may have impacted the lack of significance in the results. Because the information was relatively new for participants, it is possible that any message would increase behavioral intentions to get an endometriosis screening. Future studies might find more diverse results when using a no message control condition. Finally, because the information on endometriosis was some participants first experience learning about the disease, this may explain the lack of variance in reported self-efficacy. For unknown illness or lack of awareness surrounding a disease, measuring and testing awareness rather than self-efficacy may be more effective in understanding message effects.

## 6. Conclusions

This study contributes to the current literature in several ways. Firstly, much is unknown about young women’s awareness and screening behaviors related to endometriosis. This contrasts with the larger global population of women who are plagued with the disease, which may be inaccurate due to the normalization and downplaying of women’s pain associated with menstruation. By examining this topic, we can better understand the roles narrative/non-narrative formats, as well as emotional appeals, play in increasing women’s intentions to get screened for endometriosis. Creating awareness surrounding this disease can impact an individual’s life for the better if they currently suffer from endometriosis. Second, the research not only expands the research on hope as a useful emotional appeal that can be implemented in message design for an under-studied health issue but also expands the growing literature surrounding emotional appeals. The focus of both narrative persuasion research and health communication research, in general, has been on negative emotional appeals for decades [66,67,68,69], while few studies have shown that positive and meaningful emotions can be just as effective, if not more effective, in impacting behavioral intentions toward behavioral change. Because hope can be a complex emotion to employ in narratives, this study contributes to the body of research that informs the creation and implementation of narrative messages that incorporate elements of hope.

## Figures and Tables

**Table 1 ijerph-20-06209-t001:** Participant Demographics.

Sample Characteristics	*n*	%	*M* (*SD*)
Gender			
Male ^a^	1	0.3	
Female	325	99.7	
Race			
White	253	77.6	
Black or African American	11	3.4	
American Indian or Alaskan Native	2	0.6	
Asian	36	10.9	
Multiracial	19	5.8	
Other Race	5	1.5	
Ethnicity			
Non-Hispanic White	304	93.3	
Mexican, Mexican American, Chicano	3	0.9	
Puerto Rican	2	0.6	
Cuban	3	0.9	
Another Hispanic, Latino, or Spanish Origin	14	4.3	
Age (range: 18–27 years old)			19.03 (1.15)
18	129	39.6	
19	105	32.3	
20	63	19.3	
≥21	29	8.8	

Note. *n* = 326. ^a^ One participant identified as biologically female and as transgender male and was therefore included in the sample.

**Table 2 ijerph-20-06209-t002:** Descriptive Statistics for Mediator and Dependent Variables by Condition.

Conditions	Efficacy		Positive Affect		Intention	
	*M*	*SD*	*M*	*SD*	*M*	*SD*
Non-Narrative	4.25	0.57	1.62	0.82	2.82	0.87
Fear Narrative	4.19	0.62	1.71	0.74	2.94	0.89
Hope Narrative	4.15	0.86	2.30	0.86	2.75	1.10

**Table 3 ijerph-20-06209-t003:** Bivariate Correlations among Focal Variables.

Variables	1	2	3	4
1. Efficacy				
2. Positive Affect	−0.08			
3. Readiness	0.11 *	0.23 **		
4. Intention	0.16 *	0.41 **	0.45 **	

Note: * *p* < 0.05. ** *p* < 0.01.

**Table 4 ijerph-20-06209-t004:** Summary of Direct and Indirect Effects (H1, H2, and H3).

Direct Pathways	*df*	*F*	*p*	95% CI
H1: Narrative vs. Non-narrative → Intention	(1, 321)	0.07	0.80	[−0.26, 0.20]
H2a: Hope vs. Fear → Self-efficacy	(1, 215)	0.51	0.47	[−0.10, 0.22]
H2b: Hope vs. Fear → Positive Affect	(1, 215)	26.61	<0.001 ***	[−0.79, −0.35]
H2c: Hope vs. Fear → Intention	(1, 215)	1.94	0.17	[−0.08, 0.46]
Indirect Pathways	Indirect Effect			95% BC·CI
H3a: Hope vs. Fear → Self-efficacy → Intention	−0.02			[−0.09, 0.04]
H3b: Hope vs. Fear → Positive Affect → Intention	0.11			[0.01, 0.23]

Note: CI = confidence interval. BC·CI = bias corrected confidence interval. The mediation analysis results (i.e., the indirect effects) presented in the table were obtained through the use of PROCESS Model 4. Results associated with the significant direct and indirect pathways were bolded. *** *p* < 0.001.

## Data Availability

The data presented in this study are available on request from the corresponding author. The data are not publicly available due to privacy and ethical restrictions.

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
