# Peer review of "Narrative Messages and the Use of Emotional Appeals on Endometriosis Screening Intention: The Mediating Role of Positive Affect"

_ijerph, 2023, doi:10.3390/ijerph20136209_

Round 1

Reviewer 1 Report

An interesting study, however, to understand the methodology properly I would recommend providing the questionnaire.

 The main question is properly addressed and the topic is interesting and seems novel. It addresses a relatively deficient field. the methodology seems well established but to be understood in a better way it is advisable to provide the questions in a supplementary file.   The conclusion is seems consistent with the arguments presented . However, to provide a solid advice i need to read the questions used in the survey.    An input of a statistician is advised as well    I hope you find these comments useful 

Reviewer 2 Report

This manuscript presents the results of an online survey-based experiment focused on the efficacy of narrative messages vs. non-narrative messages for promoting endometriosis screening intention, as well as to evaluate the effectiveness of hope appeal vs. fear appeal in narrative messages. Participants were a convenience sample of 18–30-year-old young women (N = 326).

The most positive aspects of the present study are related to the following questions:

1.- From a theoretical point of view, this work contributes to the analysis of the explanatory mechanisms of narrative health messages to raise awareness of health issues, in particular endometriosis.

2.- The present work is based on an experimental design that allows two comparisons: a) the effectiveness of narrative messages (testimonial type) versus non-narrative messages (that deliver statistical or abstract information); and, b) the effectiveness of the emotional narrative based on hope versus that based on fear.

3.- The theoretical review in the introduction is comprehensive and focuses on the main works on narrative persuasion and health that are relevant in the context of the study.

4.- It is particularly interesting to note that hope-based narratives can enhance communication strategies for designing health campaigns to promote endometriosis prevention. This connects with previous research on restorative narratives and also on inspiring media (see and cite, Oliver et al, 2021). 

5.- It is also positive to include in the discussion the role on individual differences as mediating variables and, in particular, the stage of change in which people find themselves when faced with a given health problem, taking the Transtheoretical Model of Behavior Change as a reference.

6.- At the methodological level, the study presents the results of an online experiment that used a three-group randomized design to test three hypotheses and a research question. The design used is adequate to test these predictions.

Despite the merits of the manuscript, some issues have been identified that authors should address:

1.- The expression "full mediation" (p. 3, line 115) should not be used as it is outdated and has been widely criticized by experts in mediation analysis (see Hayes, 2022).

2.- The introduction mentions the possible positive effect of hope-based narratives, but does not mention any studies on the effect of fear-based narratives. Since both strategies (hope and fear) are compared in this research, it would be positive to mention some study on this topic and connect with a clearer rationale about why hope is expected to have a greater impact than fear. There is a whole line of research on fear appeals that is not mentioned and that is relevant.

3.- On page 4 (line 192) the word “hypothesis” (in singular) should be used.

4.- It should be clarified why self-efficacy is considered to be a mediating mechanism and not an outcome variable.

5.- The heading "Manipulation & Attention Checks" (page 6, line 298) should start in another paragraph.

6.- There are two important issues about the "Positive Affective Response" measure that should be clarified. First of all, what was the statement to assess the emotional responses of the participants? Were the participants asked about the emotions felt while reading the message or were they asked about the emotions felt "in these moments", immediately after reading the message? Secondly, it is striking that the mean in the indicator of positive emotions was so low, even in the participants exposed to the narrative message based on hope. How do the authors explain this result? By the way, the average in the behavioral intention indicator (participant's endometriosis screening intention) was also very low.

7.- The description of the "Stages of Change" measure should appear in the manuscript before the description of the "Self-Efficacy" measure.

8.- Table 4 should be deleted and replaced by a table that integrates the results of the mediation analyzes with PROCESS (model 4): indirect effects. By the way, when reporting the results of the mediation analysis, the expression “effect size” should not be used, but rather “indirect effect”.

9.- Since the independent variable is a multicategorical variable, it should be reported in more detail in the manuscript how said variable was coded. Assuming that the “reference group” was the non-narrative message (value 1), the fear-based narrative message was the second condition (value 3), and the hope-based narrative message was the third condition (value 3), the statistical approximation to code the independent variable should be the Helmert coding system (see Hayes, 2022). Using the Helmert coding system, PROCESS creates two variables (X1 and X2) which allows quantifying the “specific relative indirect effects” of the independent variable on the outcome variable through the two mediating mechanisms proposed in this study (self-efficacy and positive affect). X1 compares the effect of the narrative messages (taken together) versus the non-narrative message. And X2 compares the effect of hope-based narrative message versus fear-based narrative message.

10.- It is recommended to include a figure with the results of the mediational analysis (H3) to show the different regression coefficients (not standardized) on the different paths of the model.

References

Hayes, A. F. (2022). Introduction to mediation, moderation, and conditional process analysis: A regression-based approach (3rd edition). The Guilford Press.

Oliver, M. B., Raney, A. A., Bartsch, A., Janicke-Bowles, S., Appel, M., & Dale, K. (2021). Model of inspiring media. Journal of Media Psychology33(4), 191-201. https://doi.org/10.1027/1864-1105/a000305
